# Impacts of a Profile Failure of the Cycloidal Drive of a Planetary Gear on Transmission Gear

**Attila Csobán** 

Department of Machine and Product Design, Faculty of Mechanical Engineering,
Budapest University of Technology and Economics, Műegyetem rkp. 3, 1111 Budapest, Hungary;
csoban.attila@gt3.bme.hu

**Abstract:** Recently, cycloidal drives have been increasingly used due to their beneficial features, such as the implementation of a large transmission, efficiency, and high performance density. Production accuracy is inevitable in order to ensure the dynamically proper and smooth operation of the drive. Smaller backlash and minimal transmission fluctuations can only be achieved by improving the production accuracy, reducing the number of production failures, and shrinking the tolerance zone. This research primarily focuses on the investigation of the tolerable production accuracy of small-series and individually manufactured drives. The analysis of load distribution was calculated on the cogs of a planetary gear made with a wire EDM machine. On the other hand, we investigate how production failures affect transmission fluctuations and the backlash of a drive. The novelty of this research is based on the determined analytical equations, which can help engineers to find the right tolerances to a given gear ratio fluctuation.

**Keywords:** profile failure; production failures; transmission fluctuation; cycloidal drive

## 1. Introduction

The investigation of the impact of production failures on kinematic features was performed by supposing different levels of production accuracy and tolerance zones created on different cog geometries [1–4]. Based on the findings of this study, we can determine the required minimum accuracy level for a particular cog geometry, which ensures achieving the predefined maximum level of transmission fluctuation [5–8].

When performing the calculations, geometrical equations describing the connection of cogs were solved by numerical methods using a special software developed for this purpose. The grade of production failures was determined according to particular tolerance zones both in the load distribution and the definition of kinematic features.

Transmission fluctuation caused by production failures was established numerically for different gear geometries. The findings suggest that the transmission changes depending on the rotation angle of the output shafts [5–8]. The required production accuracy can be estimated for the particular maximum transmission failure in advance by relying on a derived analytical formula based on the maximum transmission fluctuation calculated for a particular module, profile failure, and cog number.

The manuscript's main sections are as follows:

- Production failures: The profile failures of the cycloidal planetary gears are discussed.
- Definition of the impact of the profile failure on the transmission: The effect of the cycloidal planetary gear profile failure on the transmission is discussed.
- Examined gear geometries: The analyzed cycloidal gear geometries are presented.
- Results: Our obtained results are presented.
- Discussion and Conclusions: Summary of the research, with the major results highlighted.

## 2. Production Failures

In the main components of the drives with cycloidal cogs, diameter failures of the pitch-circle may occur [9,10]. There can be distribution failures, profile failures, diameter failures of driving bores, or hit failures of the rolling circle; however, hit failures of the planetary gear are also possible due to an eccentricity failure.

When cycloidal planetary gears are produced, the following accuracy levels are applicable: IT5 and IT6 and, in the case of individually manufactured small-series drives IT7, IT8, and, in rare cases, IT9. In this research, we examine the connection between profile failure and transmission fluctuation.

During the calculations, profile failure can be generated by shifting the theoretical cog curve without any failures by value "*d*" (failure) into the normal direction of the profile. The interpretation of the profile failure can be seen in the following figure (Figure 1).

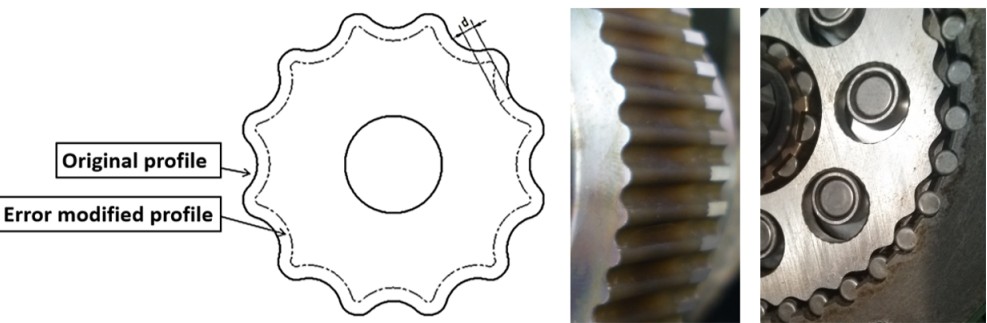

**Figure 1.** Interpretation of profile failure and the planetary gear profile.

The grade of the profile failure has been defined in due consideration of the tolerance zone determined by the diameter of the pitch-circle of the planetary gear and the particular accuracy level in advance. In Table 1, profile failures are shown.

**Table 1.** Profile failures in the case of $m = 3 [mm]$.

|    | $z_1$ | 9 | 25 | 43 | 53 | 117 |
|----|----|--------|--------|--------|--------|--------|
|    | 5 | 0.0045 | 0.0065 | 0.009 | 0.009 | 0.0125 |
|    | 6 | 0.0065 | 0.0095 | 0.0125 | 0.0125 | 0.018 |
| *IT* | 7 | 0.0105 | 0.015 | 0.02 | 0.02 | 0.0285 |
|    | 8 | 0.0165 | 0.023 | 0.0315 | 0.0315 | 0.0445 |
|    | 9 | 0.026 | 0.037 | 0.05 | 0.05 | 0.07 |

## 3. Definition of the Impact of the Profile Failure on the Transmission

During the research, the maximum deviation of the transmission from the theoretical figure was determined with different profile failures for several modules and cog numbers. First, the geometrical connections required to define the transmission figure were derived, later we examined how profile failures affect transmissions. Equations describing the connection of gears were solved using numerical methods in a Scilab environment.

Cycloidal drives are high power-density gearboxes consisting of one, but preferably two (3) planetary gears, one (4) ring gear, and one (2) eccentric shaft driving the planetary gears. The numbers between the parentheses represents the components in Figure 2. During the calculations, the initial state were determined by the theoretical curve without any failures. An interpretation of the initial state and a schematic concept of the cycloidal drive are illustrated in Figure 2.

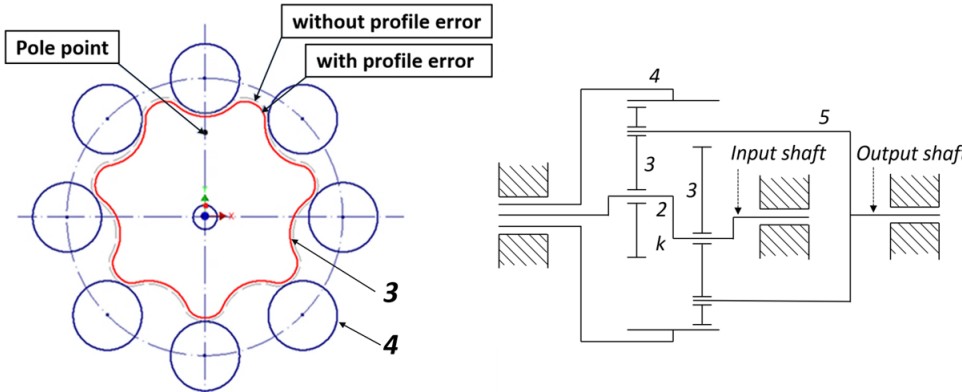

**Figure 2.** Interpretation of the initial state and on the right schematic concept of the cycloidal drive.

The first step is to define the rotation of the (2) input shaft, which is necessary for the hit ($\alpha$). The required rotation for the collision can be seen in Figure 3.

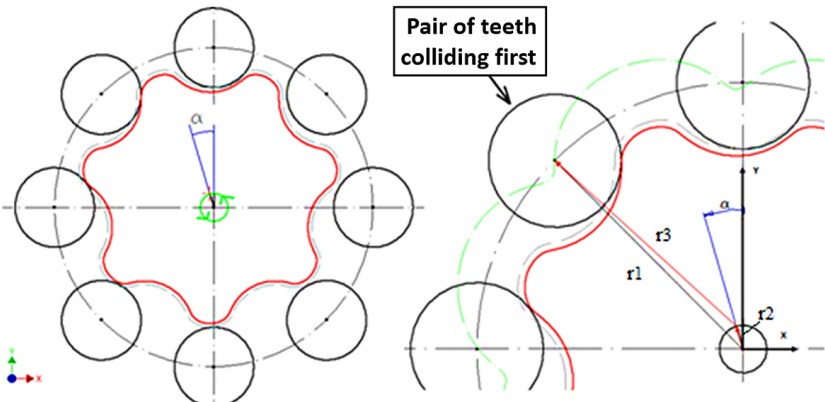

**Figure 3.** The rotation of the input shaft required for the collision ($\alpha$).

When the shaft rotates, the planetary gear makes an orbital motion, i.e., it does not rotate, but its center point moves along the circle determined by the (3) eccentric cam (Figure 3). The rotation angle of the (2) input shaft required for the hit always has to be determined for the first pin located left from the pole point (where the absolute velocity equals zero) applicable to the theoretical curve (without any failures).

The contact of two curves, i.e., the profile curve and the circle (pin curve) has to be examined. In the case of a hit, the center point of the pin will be at a radius distance from the profile curve and thus the profile curve can be shifted by the pin radius toward the normal direction of the profile. Therefore, only the contact between one curve and one point has to be examined. When the pin in question hits, the following connection can be established according to the figure:

$$r_1 - r_{2(\alpha)} - r_{3(\varphi)} = 0 \tag{1}$$

where $r_1$ is the vector pointing at the center point of the pin in question, $r_{2(\alpha)}$ is the vector pointing at the center point of the planetary gear, $\alpha$ is the rotation angle that is necessary for the eccentric cam collision, $r_{3(\varphi)}$ is the vector describing the profile of the planetary gear shifted by the pin radius ($R_{cs}$) into the normal direction, and $\varphi$ is the parameter of the equation depicting the curve.

Therefore, we can find the figures of $\alpha$ and $\varphi$ to which the above-mentioned equation applies, i.e., where the profile shifted by the radius $R_{cs}$ crosses the center point of the pin, the next step is to determine the gap between the cog pairs. To determine the gap between the cog pairs, see the following figure (Figure 4):

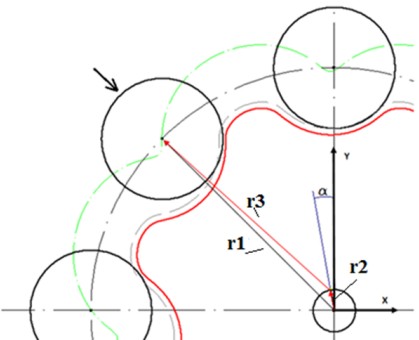

**Figure 4.** Determination of the gap.

In the case of a particular cogpair, the following equation can be established to define the gap:

$$r_1 - r_{2(\varphi_1)} - r_{3(R,\varphi)} = 0 \tag{2}$$

where $r_1$ is the vector pointing at the center point of the pin in question, $r_{2(\varphi_1)}$ is the vector pointing at the center point of the planetary gear, $\varphi_1$ is the angle determining the location of the (3) eccentric cam, $r_{3(R,\varphi)}$ is the vector describing the shifted cycloidal curve, and $R$ is the value by which the interrupted curve is shifted into the normal direction of the profile.

The calculation is performed in the above-mentioned way with the difference that now $R$ is one of the variables (unknown parameters); therefore, the gap is:

$$s = R - R_{cs} = 0 \tag{3}$$

where $R_{cs}$-is the pin radius.

The next step is to determine the transmission. To do that, it is necessary to determine the rotation angle of the (5) output shaft ($\varphi_2$) depending on the rotation angle of the (2) input shaft ($\varphi_1$). The definition of the rotation angle can be seen in Figure 5.

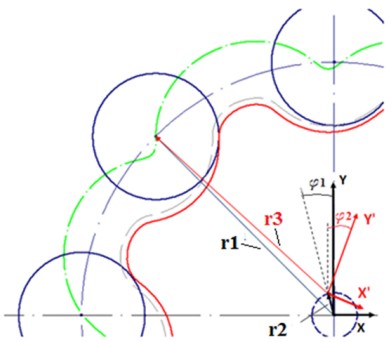

**Figure 5.** Definition of the rotation angle of the planetary gear.

During the motion of the planetary gear, the curve shifted by the radius from its profile always crosses the center point of the connected pin.

The equation describing the shifted curve is given in the coordinate system $X'Y'$, which is attached to the planetary gear. The vector indicated in this system ($r_3$) will be converted into the coordinate system $(X, Y)$ created for the (4) ring gear. After that, we make it equal with the vector pointing at the center point of the pin ($r_1$).

When two pins are connected, the following equation shall be applicable using the signs of the figure:

$$r_1 = T \cdot r_{3(\varphi)} + r_{2(\varphi_1)} \tag{4}$$

$$T = \begin{bmatrix} \cos(\varphi_2) & \sin(\varphi_2) \\ -\sin(\varphi_2) & \cos(\varphi_2) \end{bmatrix} \tag{5}$$

where $r_1$ is the vector pointing at the centerpoint of the connected pin, $T$ is the transformation matrix, $r_{3(\varphi)}$) is the vector containing the parameter equations describing the shifted curve in the coordinate system $X'Y'$ where $\varphi$ is the parameter of the equations describing the curve, $r_{2(\varphi_1)}$ is the vector pointing at the centerpoint of the planetary gear, $\varphi_1$ is the rotation angle of the (2) input shaft, and $\varphi_2(\varphi_1)$ is the rotation angle of the planetary gear.

The two variables $\varphi$ and $\varphi_2$ can be determined for the figure $\varphi_1$ as indicated in the above-mentioned equation. Knowing the figures $\varphi_2$ and $\varphi_1$, the transmission can be calculated as follows (Figure 6):

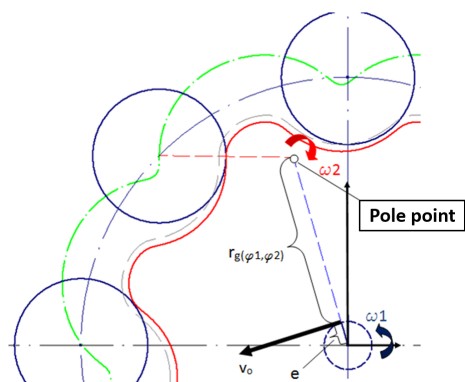

**Figure 6.** Determination of the gear ratio.

If the rotation of the planetary gear ($\varphi_2$) aligned to the particular rotation value of the (3) eccentric cam ($\varphi_1$) is known, the location of the pole point can be determined based on these values (Figure 6).

The speed of the center point of the planetary gear can be described in two ways:

$$v_0 = \omega_1 \cdot e \tag{6}$$

$$v_0 = \omega_2 \cdot r_{g(\varphi_1, \varphi_2)} \tag{7}$$

Equations (6) and (7) are made equal and arranged:

$$i = \frac{\omega_1}{\omega_2} = \frac{r_{g(\varphi_1, \varphi_2)}}{e} \tag{8}$$

where $e$ is the eccentricity (axial distance) and $r_{g(\varphi_1, \varphi_2)}$ is the distance of the center point of the planetary gear from the pole point.

The above-mentioned equations are solved numerically. The process diagram of the algorithm is shown in the following figure (Figure 7).

Here, $\alpha$ is the rotation angle of the (3) eccentric cam required for the collosion, $N$ is the number of the connected pins, $s_i$ is the gap calculated at the pin i, $N_{(min(si))}$ is the number of pins defined for the minimum gap, $\varphi_1$ is the rotation angle of the (3) eccentric cam ($\varphi_1 = \varphi_1 \ldots \varphi_n$), $\varphi_2$ is the rotation angle of the planetary gear, and $\Delta\varphi_1$ is the grade of the rotation angle of the (3) eccentric cam.

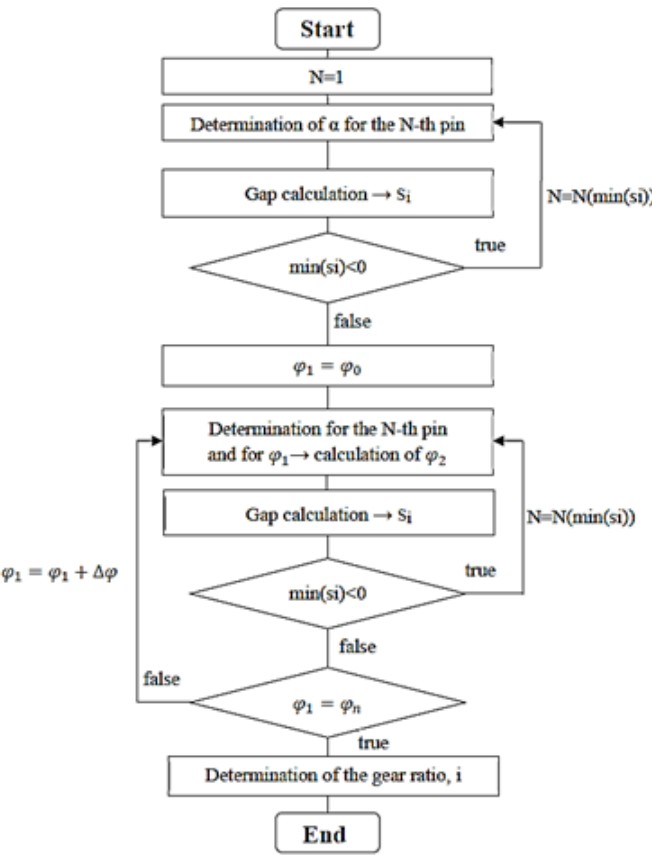

**Figure 7.** Flow chart of the calculation process.

We use the data of the examined gear pairs of a particular drive as an example. You can see the parameters of the examined gear pairs in the following Table 2:

**Table 2.** Gear parameters.

| | |
|---|---|
| Module $(m)$ : | $1[mm]$ |
| Number of teeth planet-gear $(z_1)$ : | $9[-]$ |
| Number of teeth ring-gear $(z_2 = z_1 + 1)$ : | $10[-]$ |
| Addendum modification $(x)$: | $0.3[-]$ |
| Generating circle radius factor $(r_c^*)$: | $1[-]$ |
| Profile failure $(d)$ : | $0.1[mm]$ |

As it is shown by the algorithm, the rotation angle of the (2) input shaft needed for the hit was determined first. It can be seen that pin number 2 hits first (Figure 8). On the left-hand side of the following image, the gap belonging to certain pins are marked with a circle. The gap size is proportional to the circle diameter. On the right-hand side of the picture, the column diagram also shows the gap size.

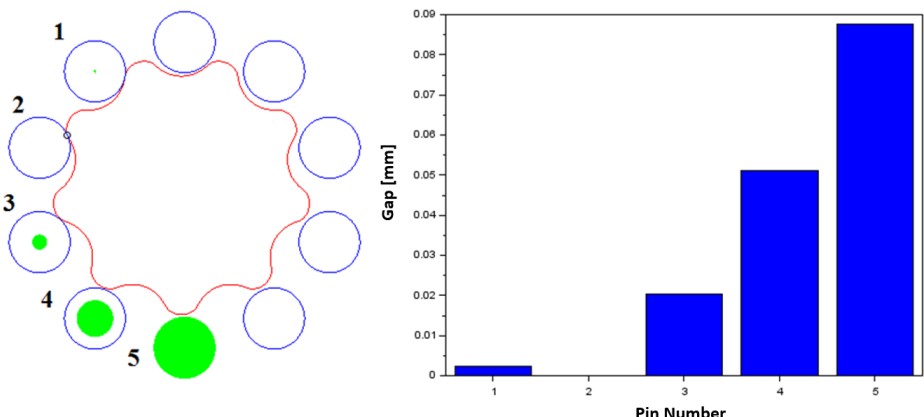

**Figure 8.** Gaps in the case of certain cog-pairs.

The transmission was determined supposing the rotation of the (3) eccentric cam in $360[°]$ $(\varphi_1)$, in grades of $2[°]$ $(\Delta\varphi_1)$. Rotation angle figures of the planetary gear aligned to particular rotation angles of the (3) eccentric cam are shown in the following figure (Figure 9).

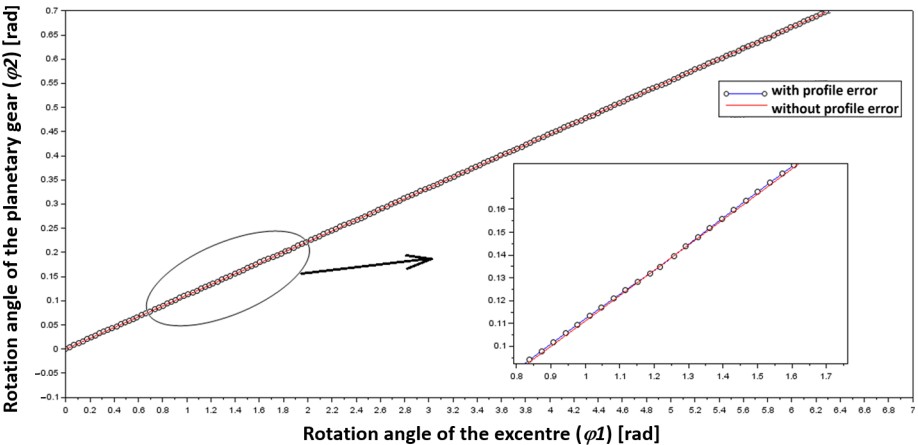

**Figure 9.** Rotation angle of the planetary gear $(\varphi_2)$ depending on the rotation angle of the (3) eccentric cam $(\varphi_1)$.

Calculation results are marked with dotted lines, and cases without any failures are marked with continuous lines in the picture. The latter one is a straight line because the transmission is constant. It can be seen in the enlarged segment that they are not identical, which is due to the profile failure.

Knowing the rotation angle of the planetary gear as-signed to the particular rotation angle of the (3) eccentric cam, the transmission change can be established using the equation shown above (8) or through derivation since the following equation applies to both rotation angles.

As shown by the algorithm, the rotation angle of the (2) input shaft needed for the hit was determined first. It can be seen that pin number 2 hits first (Figure 9). On the left-hand side of the following image, the gap belonging to certain pins are marked with a circle. The gap size is proportional to the circle diameter. On the right-hand side of the picture, the column diagram also shows the gap size.

$$i = \frac{d\varphi_1}{d\varphi_2} \tag{9}$$

This indicates that the transmission is the rise of the curve marked with dotted lines compared to the vertical axis. In this case, this means a numerical derivation according to the following equation:

$$i_n \cong \frac{d\varphi_{1(n+1)} - d\varphi_{1(n-1)}}{d\varphi_{2(n+1)} - d\varphi_{2(n-1)}} \tag{10}$$

The following diagram shows the transmission change, which was determined in two different ways. The transmission change calculated using Equation (9) is marked with $i_1$, and the change in transmission determined using Equation (10) is marked with $i_2$. The transmission was determined supposing the rotation of the (3) eccentric cam in $360[°]$ ($\varphi_1$), in grades of $2[°]$ ($\Delta\varphi_1$). Rotation angle figures of the planetary gear aligned to particular rotation angles of the (3) eccentric cam are shown in the following figure (Figure 10).

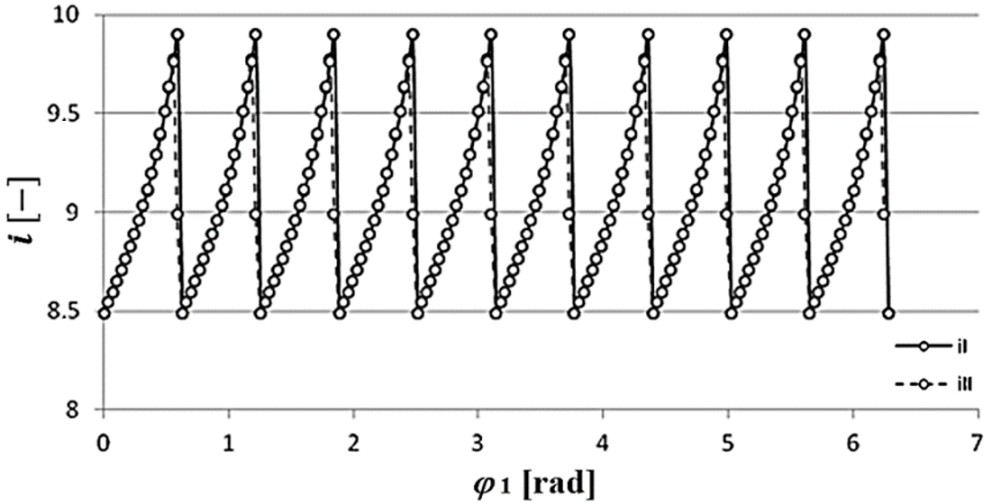

**Figure 10.** Transmission change and fluctuation.

It is clearly seen that transmission is not constant, it fluctuates around the value without any failures ($i = 9$). Furthermore, the curve describing the transmission change is not continuous. The reason for that is that there is only one cog-pair connection because of the failure, and the connected cog-pairs vary during the movement of the planetary gear. Where the curve is interrupted, cog-pairs leave the connection, and the next cog-pair will determine the motion. Therefore, each segment of the curve belongs to different cog-pairs.

Transmission fluctuations determined in two different ways have brought approximately the same results. The only difference is that the transmission ($i2$) determined by Equation (10) does not bring the real result at the points of the function $\varphi_2 = f(\varphi_1)$ where the derived figure can not be construed. These points are the maximum points of the above diagram; therefore, it is advisable to use Equation (11) in the future.

The percentile deviation of the real transmission from the original one can be determined by the following equation:

$$\Delta i = \frac{i_{ideal} - i_{real}}{i_{ideal}} \cdot 100 \tag{11}$$

The deviation calculated by using Equation (10) is shown in the next figure (Figure 11).



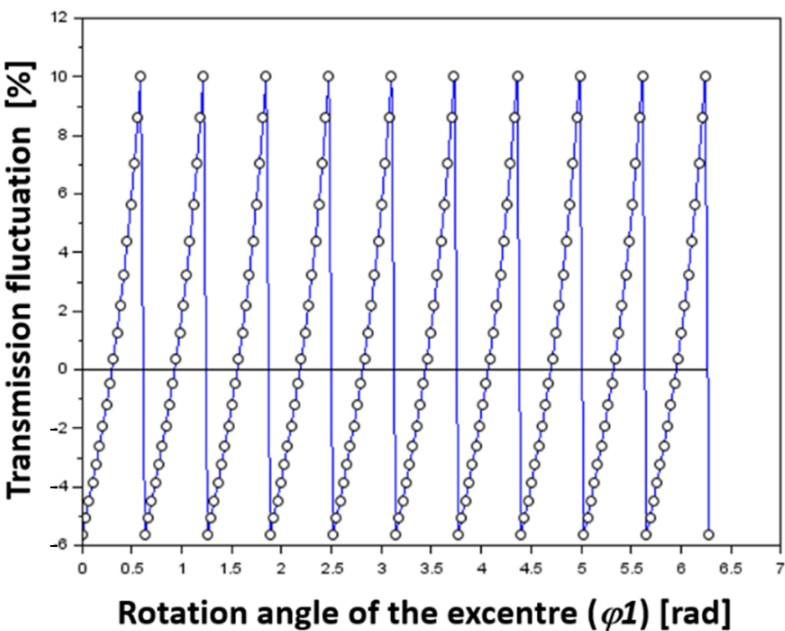

**Figure 11.** Percentile deviation of the transmission.

It is important to investigate how further cog parameters affect transmission fluctuation. The percentile transmission fluctuation calculated by using Equation (10) is shown in the following examples. The impact of the increase of the profile shift coefficient is shown in the next figure (Figure 12).

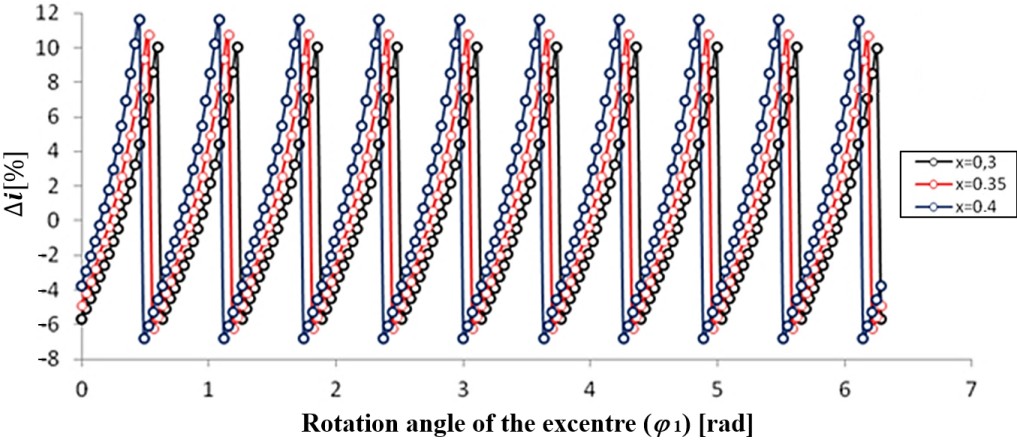

**Figure 12.** Impact of the profile shift coefficient ($x$) on transmission fluctuation ($m = 1; z_1 = 9; d = 0.1$).

By increasing the profile shift coefficient, the maximum value of transmission fluctuation can be raised. The impact of profile failures on transmission fluctuation is shown in the next diagram (Figure 13).

Reduction of the number of profile failures brings the expected result, meaning the transmission fluctuation decreases. The fluctuation decreases at the rate of failure reduction (approximately), i.e., if the failure is reduced by 90% (it decreases from $0.1 [mm]$ to $0.01 [mm]$), the maximum value of fluctuation also decreases to its tenth.

Curves describing the fluctuation show that the curve will develop as many sections ($z_2$) during the complete rotation that suit the cog number of the (4) ring gear. These sections coincide with each other. For this reason, the maximum fluctuation does not need to be determined in consideration of the complete rotation, it is enough to take only one cog-pair into account.

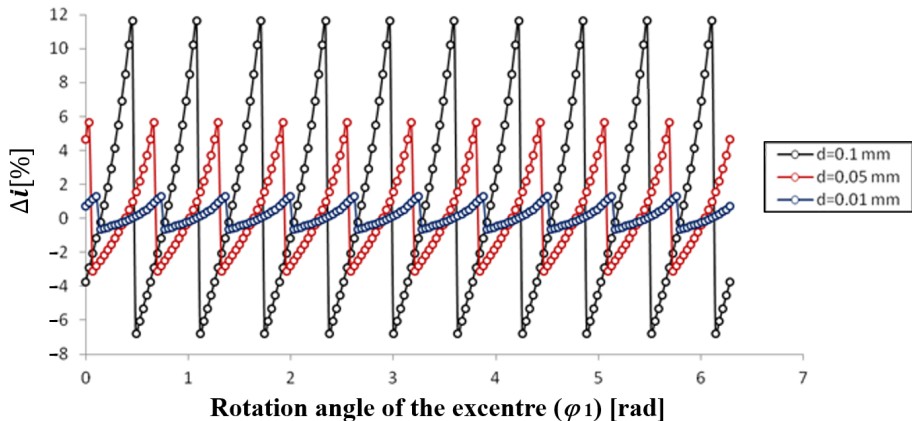

**Figure 13.** Impact of profile failures ($d$) on the transmission fluctuation ($m = 1; z_1 = 9; x = 0.3$).

In the case of the same profile failures, the transmission of any change in the cog number modifies the transmission as follows (Figure 14).

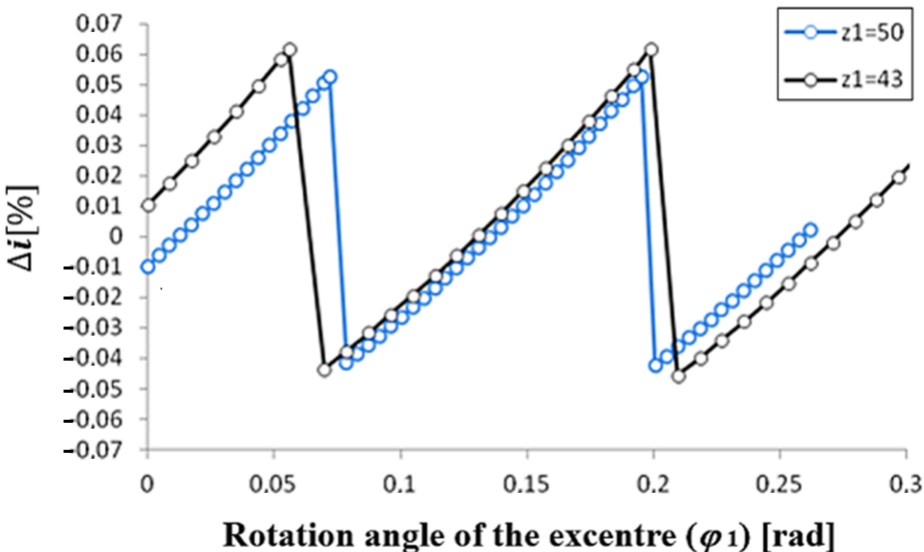

**Figure 14.** The impact of different gear teeth with a constant profile error on the transmission fluctuation ($d = 0.01; x = 0.3$).

## 4. Examined Gear Geometries

The diameter of the pitch-circle is determined by a module and the cog number. Based on the indicated cog number and module, this is 100 different gear sizes; however, in the case of $m = 5[mm]$ and $z1 = 117[-]$, the diameter of the pitch-circle is larger than $500[mm]$ (this is a rare application in practice), and thus calculations are only performed for 95 different gear geometries.

The accuracy levels used to define profile failures are as follows: IT5, IT6, IT7, IT8, and IT9. You can see the examined gear parameter in Table 3.

**Table 3.** Gear parameters.

| Module range ($m$) : | $1, 2, 3, 5[mm]$ |
|---|---|
| Pinion teeth range ($z_1$) : | $9, 25, 43, 53, 117[-]$ |
| Addendum modification ($x$): | $0.3[-]$ |
| Generating circle radius factor ($r_c^*$): | $1[-]$ |

## 5. Results

All the given analytical results can be used in the design process. The resulting findings of the research are shown in the following Figures 15–18.

Results belonging to the value $m = 1[mm]$ :

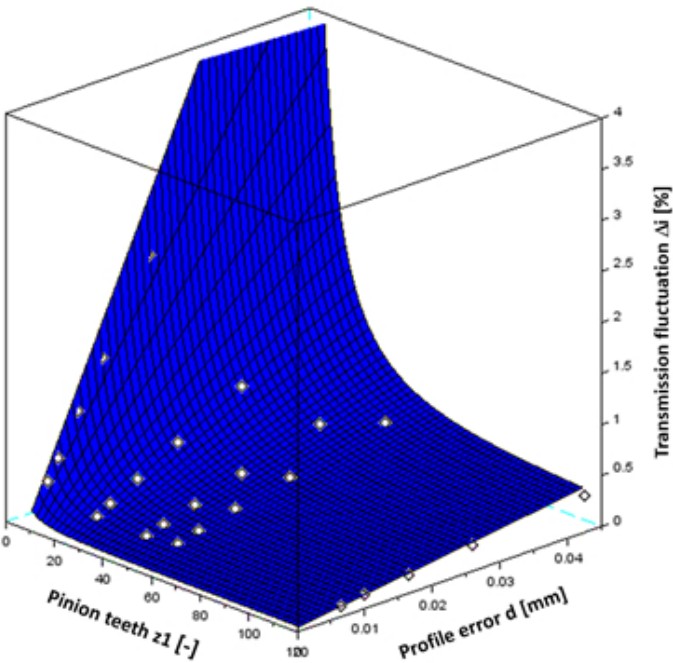

**Figure 15.** Maximum transmission fluctuation in the case of $m = 1[mm]$.

Equation for the fitted surface:

$$\Delta i = 6.842 \cdot d + 3351.9 \cdot d \cdot z_1^{-1.5} \tag{12}$$

where $d$ is the profile error, and $z_1$ is pinion teeth.

Results belonging to the value $m = 2[mm]$ :

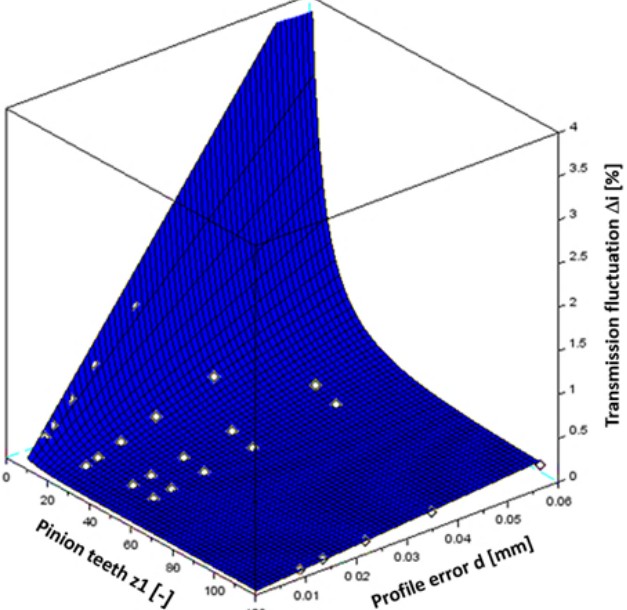

**Figure 16.** Maximum transmission fluctuation in the case of $m = 2[mm]$.

Equation for the fitted surface:

$$\Delta i = 3.464 \cdot d + 1720.82 \cdot d \cdot z_1^{-1.5} \tag{13}$$

Results belonging to the value $m = 3[mm]$ :

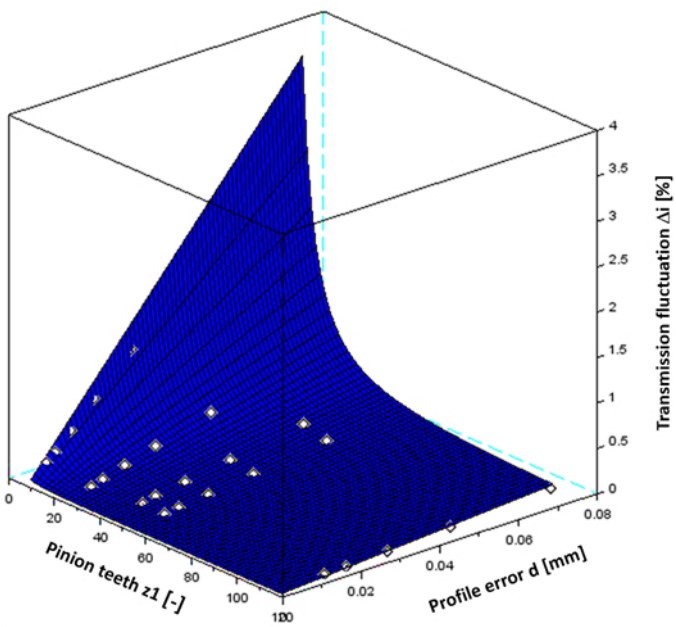

**Figure 17.** Maximum transmission fluctuation in the case of $m = 3[mm]$.

Equation for the fitted surface:

$$\Delta i = 2.293 \cdot d + 1156.47 \cdot d \cdot z_1^{-1.5} \tag{14}$$

Results belonging to the value $m = 5[mm]$ :

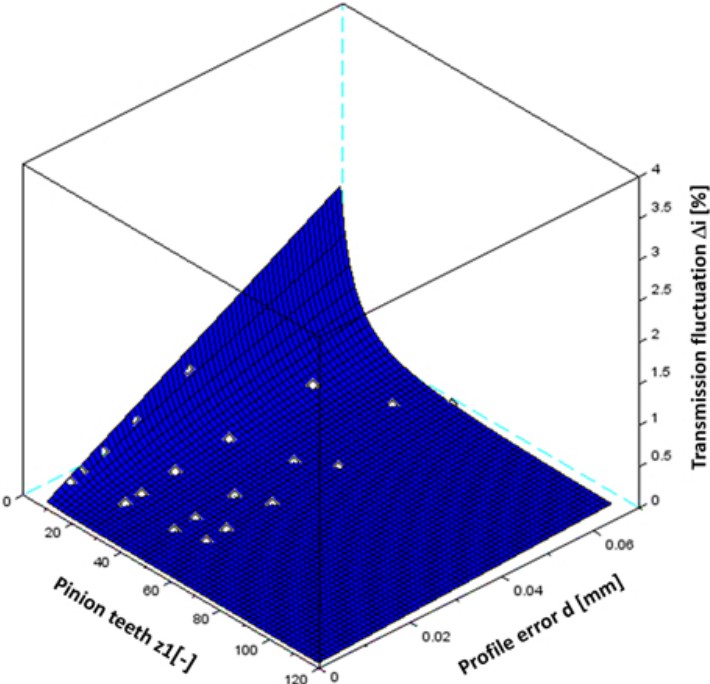

**Figure 18.** Maximum transmission fluctuation in the case of $m = 5[mm]$.

Equation for the fitted surface:

$$\Delta i = 1.838 \cdot d + 680.33 \cdot d \cdot z_1^{-1.5} \tag{15}$$

The results and the equation of the fitted surface also suggest that the maximum value of the transmission failure changes linearly depending on profile failures and along a hyperbole in terms of the cog number. Knowing the approximate connections (equations), the minimum production accuracy can be estimated if we know the maximum transmission fluctuation as well.

## 6. Discussion and Conclusions

The results and the equation of the fitted surface also suggest that the maximum value of the transmission failure changes linearly depending on profile failures and along a hyperbole in terms of the cog number. The minimum production accuracy can be estimated if we know the maximum transmission fluctuation. Using the method of the smallest squares, the analytical equation of the transmission fluctuation can be created when fitting a surface to the points representing the results. The percentile deviation of the real transmission from the original one can be determined with the derived equation.

**Funding:** The research reported in this paper and carried out at BME was supported by the NRDI Fund (TKP2020 NC, Grant No. BME-NCS) based on the charter of bolster issued by the NRDI Office under the auspices of the Ministry for Innovation and Technology.

**Acknowledgments:** Hereby, I would like to express my thanks to the engineering student Dávid Roboz who made a considerable contribution to my research by doing a great job.

**Conflicts of Interest:** The authors declare no conflict of interest.

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
