# Peer review of "Impacts of a Profile Failure of the Cycloidal Drive of a Planetary Gear on Transmission Gear"

_lubricants, doi:10.3390/lubricants9070071_

Round 1

Reviewer 1 Report

This paper discussed the relationship of transmission fluctuation and profile failure for the cycloidal drives, which is helpful to improve kinematic features of cycloidal drives. However some results and discussion are unclear. there are still many typos and inappropriate sentences. Some inappropriate discussions are also found. Overall, in the present form it cannot be accepted for publication, Authors need revise the manuscript, again.

--Please supplement more reference about the transmission fluctuation and gear failure.

--Please describe the operating principle first and give the corresponding part name in Figure in detail.

--Line 34: What is “excentrity failure”?

--What mean is the pole point in Figure 2?

--Please indicate the name in Figure in detail.

--Please give every Figure number in the corresponding text.

--What difference is the Figure 3 and Figure 4? I think they are the same figure.

--Line 110: it is false description, it should be Equation(6) and (7)┄

--The section”Discussion and Conclusions” need to be rewritten and give the detail variation relationship of transmission fluctuation and profile failure, cog number and so on.

Author Response

Dear Reviewer, 

First of all I would like to thank you for your valuable comments. I tried to correct all the main points you suggested. In attachement you will find a corrected pdf. 

I hope that it will satisfy you! Thank you for your time and work. 

Best regards, 

Attila Csobán PhD

Reviewer 2 Report

There are some rooms to improve this paper:

1)The novelty of this paper should be highlighted in the Abstract and Introduction; otherwise, it is more like a technical report.

2) Presenting citations like [1-10] should be avoided.

3) At the end of the Introduction, a brief description of the content of each section is suggested.

4) The description 'with any failures by the figure ''d'' (failure)' is confusing.

5) Please avoid the legends shown in Figure 9, 12 and 13 hiding some presented information.

6) More descriptions are required in the Results section. The way the authors used is not acceptable. 

Author Response

(The authors gave the same response as above.)

Round 2

Reviewer 1 Report

Some presentations are clear after modified, but some revisions are still required: 1) more references need to be supplemented in "Introduction" section; 2) some conclusions should be specific according to the results.

Author Response

Thank you for your kind review. In attachment you will find a corrected version of the article. 

Best regards, 

                Attila Csobán 

Reviewer 2 Report

The concerns I pointed out still exist. There is no significant improvement compared with the original manuscript.

Author Response

 Dear Reviewer,

I am sorry that you are not satisfied with the improvements.

Based on the first-round review, I made the following corrections in the manuscript.

  1. The novelty of this paper is highlighted in the Abstract and Introduction:

“The novelty of this research is based on the determined analytical equations, which can help the engineers to find the right tolerances to a given gear ratio fluctuation.”

  1. Multi citations like [1-10] is avoided. [x], [y], [z], … citation is used.
  2. A brief description of the content of each section has been made.
  3. The description 'with any failures by the figure ''d'' (failure)' is changed.
  4. The overlept informations in Figure 9, 12 and 13 are now visible.
  5. The aim of this research was to derive equations for calculating the main parameters. The results themselves are the equations presented in the results section which are still used in our industrial practice. The results are presented according to the daily industrial use, in equation form.

As you can see modification in the manuscript were made.

Please note this.

Sincerely,

      Attila Csobán

Round 3

Reviewer 2 Report

All my concerns have been addressed. I agree to accept it.